# Activated Natural Killer Cell Inoculation Alleviates Fibrotic Liver Pathology in a Carbon Tetrachloride-Induced Liver Cirrhosis Mouse Model

**DOI:** 10.3390/biomedicines11041090

**Published:** 2023-04-04

**Authors:** Ho Rim Oh, Min Kyung Ko, Daehee Son, Young Wook Ki, Shin-Il Kim, Seok-Yong Lee, Keon Wook Kang, Gi Jeong Cheon, Do Won Hwang, Hyewon Youn

**Affiliations:** 1Department of Nuclear Medicine, Seoul National University College of Medicine, Seoul 03080, Republic of Korea; 2Department of Biomedical Sciences, Seoul National University College of Medicine, Seoul 03080, Republic of Korea; 3Cancer Research Institute, Seoul National University College of Medicine, Seoul 03080, Republic of Korea; 4Research & Development Center, THERABEST, Co., Ltd., Seoul 06656, Republic of Korea; 5Cancer Imaging Center, Seoul National University Hospital, Seoul 03080, Republic of Korea

**Keywords:** liver cirrhosis, carbon tetrachloride (CCl_4_), natural killer (NK) cells, bioluminescence imaging (BLI)

## Abstract

Activated hepatic stellate cells (HSCs) play a detrimental role in liver fibrosis progression. Natural killer (NK) cells are known to selectively recognize abnormal or transformed cells via their receptor activation and induce target cell apoptosis and, therefore, can be used as a potential therapeutic strategy for liver cirrhosis. In this study, we examined the therapeutic effects of NK cells in the carbon tetrachloride (CCl_4_)-induced liver cirrhosis mouse model. NK cells were isolated from the mouse spleen and expanded in the cytokine-stimulated culture medium. Natural killer group 2, member D (NKG2D)-positive NK cells were significantly increased after a week of expansion in culture. The intravenous injection of NK cells significantly alleviated liver cirrhosis by reducing collagen deposition, HSC marker activation, and macrophage infiltration. For in vivo imaging, NK cells were isolated from codon-optimized luciferase-expressing transgenic mice. Luciferase-expressing NK cells were expanded, activated and administrated to the mouse model to track them. Bioluminescence images showed increased accumulation of the intravenously inoculated NK cells in the cirrhotic liver of the recipient mouse. In addition, we conducted QuantSeq 3′ mRNA sequencing-based transcriptomic analysis. From the transcriptomic analysis, 33 downregulated genes in the extracellular matrix (ECM) and 41 downregulated genes involved in the inflammatory response were observed in the NK cell-treated cirrhotic liver tissues from the 1532 differentially expressed genes (DEGs). This result indicated that the repetitive administration of NK cells alleviated the pathology of liver fibrosis in the CCl_4_-induced liver cirrhosis mouse model via anti-fibrotic and anti-inflammatory mechanisms. Taken together, our research demonstrated that NK cells could have therapeutic effects in a CCl_4_-induced liver cirrhosis mouse model. In particular, it was elucidated that extracellular matrix genes and inflammatory response genes, which were mainly affected after NK cell treatment, could be potential targets.

## 1. Introduction

Liver cirrhosis is a chronic end-stage liver disease accompanied by scarring tissues due to multifactorial causes such as hepatitis, alcohol abuse, fatty liver, and drugs. Histologically, liver cirrhosis is characterized by diffuse nodular regeneration, fibrogenesis/fibrosis, and a collapse of the liver structure [1,2]. The chronic inflammatory environment of the liver tissue triggers fibrotic liver tissue formation with persistent liver damage after activation of hepatic stellate cells (HSCs) [3]. Activated HSCs are one of the main components in the pathogenesis of liver fibrosis, cirrhosis, and hepatocellular carcinoma. HSCs are quiescent in the normal liver but can be activated during liver damage to a proliferative state with a myofibroblast phenotype with migratory and invasive capabilities [4,5,6]. Patients with decompensated liver cirrhosis have a poor prognosis [7]. Although various medications can be used to reduce the temporal symptoms of liver cirrhosis [7,8], its most definitive treatment is transplantation. However, the procedure is complicated, and it is difficult to find a suitable donor.

At the preclinical level, a mouse model of liver cirrhosis can be established by the hepatotoxicity of carbon tetrachloride (CCl_4_)-induced free radicals. CCl_4_ is used to dissolve non-polar compounds, such as fats and oils, and is activated by hepatic cytochrome P450 enzymes to form trichloromethyl and trichloromethyl-peroxy radicals in the body [9,10]. These free radicals induce nucleic acid mutations and lipid peroxidation and destruct polyunsaturated fatty acids, leading to hepatotoxicity, inflammation, fibrosis, cirrhosis, and hepatocellular carcinoma [9,10].

Cell therapy can be one treatment option for liver cirrhosis. Mesenchymal stem cells (MSCs) can reduce inflammation and apoptosis, increase hepatocyte regeneration, and improve liver function by regressing liver fibrosis [11]. In particular, MSCs can migrate to the liver and differentiate into hepatocytes during liver damage [12]. However, the role of specific cell types in MSCs is still poorly understood, and their side effects in promoting tumor growth are serious concerns [13].

Natural killer (NK) cells that are important for host defense and memory-like responses could also contribute to treating liver cirrhosis [14,15,16,17]. Liver-resident NK cells showed protective effects in a CCl_4_-induced liver cirrhosis mouse model and could specifically kill activated HSCs but not quiescent HSCs [18,19,20]. Also, NK cells could negatively regulate HSCs in liver cirrhosis progression [21,22,23,24].

NK cells should be sufficiently purified and expanded for in-depth studies on their role in liver cirrhosis. The spleen, the largest secondary lymphoid organ, contains red blood cells and immune cells, including NK cells, which could be one option for purifying NK cells. However, the frequency of NK cells in the spleen of a normal 2–3-month-old C57BL/6J mouse was reported to be only about 3~4% [25]. Therefore, the efficient isolation, extraction, and expansion of NK cells are important, providing the opportunity to elucidate the role of NK cells in liver cirrhosis.

Immune cells can migrate to specific locations to implement an appropriate host defense immune response. Therefore, tracking immune cells would provide critical evidence for their role. In this context, imaging techniques represent a great choice to track cells, and, in particular, reporter gene imaging can be a powerful tool to track and visualize cells in vivo [26]. A variety of imaging techniques, including fluorescence, bioluminescence, positron emission tomography (PET), single photon emission computed tomography (SPECT), and magnetic resonance imaging (MRI), can be used for tracking cells. Among them, bioluminescence imaging (BLI) is a noninvasive visualization method that can be used in various fields of in vitro and in vivo biology due to the advantages of high sensitivity, resolution, and selectivity with a low background signal and a lack of external light excitation [26]. In the case of PET or SPECT, radioactive isotope labeling is required for cell tracking, and validation of changes in cell properties may be necessary during the labeling process [27]. In the case of an MRI, since the labeling of a contrast agent such as iron oxide is also required, it is necessary to validate that there is no change in cell characteristics during the labeling process [26,27].

In this study, we elucidated that the delivery of NK cells—generated by a specific isolation and effective expansion method—to a recipient mouse with liver cirrhosis effectively ameliorated the progression of liver cirrhosis by reducing inflammation, HSC activation, and macrophage infiltration. We observed the accumulation of therapeutic NK cells in the cirrhotic livers of mice using the BLI technique. Therapeutic NK cells were prepared from the spleens of codon-optimized luciferase-expressing transgenic mice [28] and transferred to mice with CCl_4_-induced liver cirrhosis for visualization. We performed RNA sequencing, a powerful technology that enables comprehensive gene expression analysis at the transcription level using next-generation sequencing technology [29,30]. Subsequently, we compared the transcriptomes of normal and cirrhosis-induced liver tissues by QuantSeq 3′ mRNA sequencing and further investigated the differences in transcriptomes of cirrhosis-induced liver tissues after treatment with NK cells.

## 2. Materials and Methods

### 2.1. Animals

Experiments were approved by the Institutional Animal Care and Use Committee of the Seoul National University and Seoul National University Hospital (SNU-220113-5, IACUC No. 21-0034-S1A0). Mice were housed at room temperature with 40% to 60% humidity and fed standard a chow diet and water *ad libitum* in a 12 h light/dark cycle. Mice were tested for experiments after at least one week of acclimatization.

### 2.2. Isolation and Expansion of NK Cells

Seven-week-old male BALB/c mice (n = 10) were killed by cervical dislocation, and spleens were removed and crushed to obtain splenocytes. NK cells were isolated from splenocytes by negative selection using an NK cell isolation kit (110-115-818, Miltenyi Biotec, Bergisch Gladbach, North Phine-Westphalia, Germany) according to the manufacturer’s instructions [31]. Isolated NK cells were cultured at a density of 5 × 10^6^ cells/mL in T-25 flasks in Roswell Park Memorial Institute-1640 medium (Cytiva, Marlborough, MA, USA) supplemented with 10% fetal bovine serum (FBS) (Gibco, Waltham, MA, USA), 1% L-glutamine (Gibco), 1% Antibiotic-Antimycotic (Gibco), 50 µM β-mercaptoethanol (Sigma-Aldrich, St. Louis, MO, USA) and 50 ng/mL recombinant human IL-15 protein (R&D systems, Minneapolis, MN, USA). Fresh medium was added to the cells on days 4, 5, and 6. Cells were maintained in a humidified atmosphere containing 5% CO_2_ and 95% air at 37 °C.

### 2.3. Fluorescence-Activated Cell Sorting (FACS)

Cells were suspended in 1X PBS supplemented with 0.1% FBS. Cells were washed once in buffer and stained for 30 min with PE-conjugated anti-NKG2D, FITC-conjugated anti-CD3e, APC-conjugated anti-NKp46, and anti-CD19 (Biolegend, San Diego, CA, USA). After harvesting the cells, the expression of proteins was analyzed using NovoCyte flow cytometry (Agilent Technologies, Santa Clara, CA, USA) on over 10,000 cells.

### 2.4. Interferon-γ (IFN-γ) ELISA

IFN-γ cytokine was measured by the enzyme-linked immunosorbent assay (ELISA) (SIF50C, R&D systems) using the supernatant from the NK cells in 96-well U-bottom plate. Briefly, diluent assay solution was added to the collected supernatant (100 μL each) in a well of an ELISA plate and incubated for 2 h at room temperature. Plates were then washed and incubated with INF-γ conjugate (200 μL/well) for 2 h at room temperature. The reaction was then washed and incubated with substrate solution (200 μL/well) for 30 min at room temperature. Finally, the reaction was quenched with 50 μL of stop solution. The amount of antibody was measured in a Victor Nivo 3 (PerkinElmer, Waltham, MA, USA).

### 2.5. CCl_4_-Induced Live Cirrhosis in Mice

Seven-week-old male BALB/c mice were administered intraperitoneally with mineral oil (control, n = 4) or CCl_4_ (1.5 mL/kg, diluted 1:3 in mineral oil, n = 4) twice a week for 3 weeks and once a week from the 4th week (totaling 8 weeks) [32]. NK cells were injected intravenously once a week from the 4th week after CCl_4_ injection.

### 2.6. Analysis of Fibrosis

The area of liver fibrosis was quantified with Sirius red-stained slides. Paraffin-embedded liver tissues sliced in 4 μm sections and transferred to glass slides were deparaffinized and rehydrated. The slides were treated with Picro Sirius red solution (Abcam, Cambridge, UK) for 60 min and rinsed twice with 0.1% acetic acid solution and then with absolute alcohol. Slides were washed with distilled water, mounted, and scanned using the Olympus BX43 optical microscope with a CCD camera (Olympus, Tokyo, Japan). Red areas, considered fibrotic areas, were assessed by computer-assisted image analysis using MetaMorph software 7.8.10 (Universal Imaging Corporation, Bedford Hills, NY, USA) at 40× magnification. The average value of 6 randomly selected areas per sample was used as the percent area of fibrosis.

### 2.7. Immunohistochemistry

Slides of 4 μm paraffin-embedded liver tissue sections were deparaffinized and rehydrated. After washing, the slides were boiled with 10 mM sodium citrate (pH 6.0) (ICN biomedicals, Costa Mesa, CA, USA) for antigen retrieval and then treated with 0.5% Triton X-100 (Yakuri pure chemicals, Kyoto, Japan) in Tris-buffered saline (TBS) for 5 min to permeabilize for antibodies.

Antibodies against transforming growth factor-beta (TGF-β) 1 (1:50 dilution, ab92486, Abcam), alpha-smooth muscle actin (α-SMA) (1:100 dilution, ab7817, Abcam), and F4/80 (1:100 dilution, ab6640, Abcam) were diluted with TBS containing 1% BSA and applied overnight at 4 °C. Next, the slides were incubated with secondary antibodies of donkey anti-rat IgG Alexa Fluor 488 (1:500 dilution, A-21208, Thermofisher Scientific, Waltham, MA, USA), goat anti-mouse IgG Alexa Fluor 488 (1:500 dilution, A-11001, Thermofisher Scientific), and donkey anti-rabbit IgG Alexa Fluor 555 (1:500 dilution, A-31572, Thermofisher Scientific) for 1 h. Finally, the slides were DAPI stained and mounted.

For H&E staining, the slides were deparaffinized, rehydrated, and treated with Mayer’s hematoxylin (Agilent Technologies) for 20 s and washed. Next, eosin solution (Sigma-Aldrich) was added to the slides for 30 s and washed. Finally, the slides were mounted.

### 2.8. Tracking of Therapeutic NK Cells in a Mouse Model with CCl_4_-Induced Liver Cirrhosis

For NK cell tracking experiments, five-month-old male C57BL/6 albino mice were administered mineral oil (control, n = 2) or CCl_4_ (1.5 mL/kg, diluted in mineral oil, n = 3) twice a week for 3 weeks. NK cells from codon-optimized luciferase (luc)-expressing transgenic mice were isolated and expanded [28]. After inoculation of NK-luc cells (1 × 10^7^ cells/mouse, intravenous injection) into syngeneic albino mice, D-luciferin (3 mg/mouse) (Promega, Madison, WI, USA) was injected intraperitoneally as a substrate, and in vivo bioluminescence images were acquired using the IVIS Lumina II (PerkinElmer).

### 2.9. QuantSeq 3′ mRNA Sequencing

Total RNA was isolated using the TRI Reagent solution (Invitrogen, Waltham, MA, USA). RNA quality was evaluated using 2100 Bioanalyzer with RNA 6000 Nano Chip (Agilent Technologies), and RNA quantification was determined using ND-2000 spectrophotometer (Thermofisher Scientific). Control and test RNA libraries were constructed following the manufacturer’s instructions using QuantSeq 3′ mRNA-Seq library prep kit (Lexogen, Vienna, Austria). Briefly, 500 ng of total RNA was prepared, and an oligo-deoxythymidine (oligo-dT) primer containing an Illumina compatible sequence at the 5′ end was hybridized to the RNA and reverse transcribed. After degradation of the template RNA, second-strand synthesis was performed using random primers containing an Illumina-compatible linker sequence at the 5′ end. The double-stranded library was purified by using magnetic beads from reaction mixtures. Complete adapter sequences were added through an amplification process to generate clusters, and the completed library was purified from the reaction mixture. High-throughput sequencing was performed as single-end 75 sequencing using NextSeq 500 (Illumina, San Diego, CA, USA).

### 2.10. Gene Expression Data Analysis

QuantSeq 3′ mRNA-Seq reads were aligned using Bowtie2 [33]. The alignment file was used for transcript assembly, abundance estimation, and differential expression of genes (DEGs) detection. DEGs were determined based on counts from unique and multiple alignments using coverage in BEDTools [34].

### 2.11. RNA Isolation and Quantitative Real-Time PCR (qRT-PCR)

Total RNA was extracted from cell lysates using TRI Reagent solution (Invitrogen). First-strand cDNA was synthesized from 1 μg of total RNA using the TOPscript cDNA synthesis kit (EZ005S, Enzynomics, Daejeon, Korea) according to manufacturer’s instructions. Primer sequences are provided in Table 1. Quant Studio 5 Real-time PCR system (Applied Biosystems, Waltham, MA, USA) was utilized for cDNA amplification with Power SYBR^®^ Green Master Mix (Applied Biosystems) under the following conditions: 95 °C for 10 min, followed by 45 cycles of 95 °C for 15 s and 61 °C for 1 min. Expression levels of target mRNAs were analyzed using the ∆∆Ct method and were normalized to expression levels of glyceraldehyde 3-phosphate dehydrogenase (GAPDH), a housekeeping gene used as an endogenous control. The fold changes were expressed relative to the control group.

### 2.12. Statistics Analysis

Experiments were conducted in triplicate. All data are expressed as the mean ± standard deviation (SD). Statistical significance was analyzed by *t*-test using GraphPad Prism software 9 (GraphPad Software Inc., San Diego, CA, USA). All *p*-values reported are two-sided, and significance was set at *p* < 0.05. * *p*-values < 0.05, ** *p*-values < 0.005, *** *p*-values < 0.001, and **** *p*-values < 0.0001.

## 3. Results

### 3.1. Isolation, Expansion, and Characterization of NK Cells

We chose repetitive inoculation of NK cells for the treatment of liver cirrhosis, an irreversible chronic liver disease. Therefore, a large number of mouse NK cells were required for the repetitive inoculation of NK cells to the liver cirrhosis mouse model. We carried out the negative selection of NK cells from mouse splenocytes and obtained a sufficient number of NK cells after 7 days of expansion in an appropriate medium containing cytokines (Figure 1A). Flow cytometry was used to verify the identity and purity of the isolated NK cells, and the expanded cell population was sorted and analyzed using the NK cell-specific marker NKp46. CD3e-NKp46+ NK cells, CD3e+ or CD19+ cells, and NKp46+NKG2D+ NK cells were >98%, <2%, and 61.55 ± 3.47%, respectively, in the expanded population, indicating the purity of the NK cells (Figure 1B). Moreover, on day 7, we obtained about 13.55 ± 2.40-fold NK cells compared to day 0, without any significant difference in cell viability (97.18 ± 4.46% and 94.92 ± 1.98% on day 0 and day 7, respectively) (Figure 1C). Finally, we identified significantly higher levels of IFN-γ secretion in the day 7 cells compared to in the day 1 cells (Figure 1D).

### 3.2. Significant Decrease in Collagen Deposition after Repetitive NK Cell Inoculation in the CCl_4_-Induced Liver Cirrhosis Mouse Model

Liver tissues were analyzed 3 weeks after the first intraperitoneal injection of CCl_4_ to determine the pathological features. Mice were injected with CCl_4_ twice a week for 3 weeks (Figure 2A), and liver tissues were harvested 3 weeks after the first CCl_4_ injection to verify cirrhosis. The tissue specimens were stained for H&E (Figure 2B) and Sirius red-based collagen levels (Figure 2C), showing that the liver specimens from the mineral-oil-treated control group were intact, whereas those of the CCl_4_-treated group had high collagen deposition with liver fibrotic lesions. Based on these results, the CCl_4_-treated mice could be used for further in vivo studies.

The NK cells were injected intravenously once a week from 3 weeks after the first CCl_4_ injection into CCl_4_-induced liver cirrhosis mice to verify the efficacy of the NK cell therapy (Figure 3A). After a total of six injections of NK cells (1 × 10^6^ cells each), liver tissues were harvested, and Sirius red staining was performed. Repetitive NK cell inoculations reduced fibrotic area dramatically in the liver sections (Figure 3B). The liver tissues from the CCl_4_-only treatment group showed more fibrotic areas than those from the CCl_4_ + NK cell treatment group (Figure 3B). Quantification of the fibrotic areas demonstrated that the CCl_4_ + NK cell treatment group had a significantly less area of fibrosis than the CCl_4_-only treatment group, which was similar to the liver of the control group (Figure 3B).

### 3.3. Suppression of Fibrosis after NK Cell Therapy

We further investigated whether delivered NK cells may downregulate inflammation in the liver, a possible mechanism of cirrhosis. Immunofluorescence was performed to evaluate the association of NK cell therapy with activated macrophages and hepatic stellate cells. The level of TGF-β1 secreted from the activated macrophages in the liver was higher in the CCl_4_-only treatment group than in the normal liver (Figure 4A). Alpha-smooth muscle actin (α-SMA) levels, known to be activated in HSCs, and F4/80, a macrophage infiltration marker, were increased in the CCl_4_-only treatment group (Figure 4B,C). In contrast, significant reductions of TGF-β1 (Figure 4A), activated HSCs (Figure 4B), and macrophage infiltration (Figure 4C) were observed in the CCl_4_ + NK cell treatment group. Thus, NK cell therapy could reduce the degree of fibrosis with decreased inflammation.

### 3.4. Optical Image-Based In Vivo Tracking of NK Cells Derived from Codon-Optimized Luciferase-Expressing Transgenic Mice in the CCl_4_-Induced Liver Cirrhosis Mouse Model

BLI was used to track the cells, as tracing the inoculated NK cells may be important to elucidate their site-specific roles. However, very low infection efficacy of NK cells using luciferase lentivirus is typically seen due to the intrinsic resistance of NK cells to external virus infection. Thus, we used transgenic mice stably expressing the codon-optimized luciferase (luc) gene in all tissues to obtain the NK-luc cells. After harvesting the spleens of the codon-optimized luciferase transgenic mice, NK-luc cells were isolated and expanded for 7 days (Figure 5A). The cell number-dependent luminescence signal was evaluated in splenocytes from the codon-optimized luciferase transgenic mice and isolated NK and NK-luc cells using an in vitro luminescence assay (Figure 5B,C). NK-luc cells, showing a higher luminescence signal than normal NK cells (Figure 5C), were injected intravenously into the mice (Figure 5D,E). BLI results showed that the inoculated NK-luc cells remained in the liver in the CCl_4_ + NK-luc cell treatment group compared with the control + NK-luc cell treatment group (Figure 5D,E). When organs were harvested after imaging at 36 h, the livers of the CCl_4_ + NK-luc cell treatment group showed a higher luminescence signal than those of the control + NK-luc cell treatment group. The bioluminescence signals in the other organs, with the exception of the lungs, were low in both groups. These results illustrated that the inoculated NK cells resided in the cirrhotic liver and were involved in reducing the progression of cirrhosis.

### 3.5. Gene Expression Profiling of Liver Cirrhosis Mouse Model after NK Cell Therapy

We used QuantSeq technology as an alternative to microarrays, and we used conventional RNA-seq to examine the gene expression profile of the NK cell-treated liver cirrhosis mouse model. QuantSeq 3′ mRNA sequencing was performed on the saline, CCl_4_, and CCl_4_ + NK cell-treated (CCl_4_ + NK) groups. Figure 6A shows significantly up- or down-regulated gene expression profiles, as determined by cluster heatmap analysis. DEGs were identified as genes up- or down-regulated with a ± 1.5-fold change (log2 = 1). As summarized in the expression plots, we identified 295 and 1034 of up- and down-regulated DEGs in the CCl_4_ compared with the saline group and 319 and 176 of up- and down-regulated DEGs in CCl_4_ + NK cell-treated group compared with the CCl_4_ group (Figure 6B).

The gene ontology (GO)-based bioinformatics analysis of the DEGs of the CCl_4_/saline and CCl_4_ + NK/CCl_4_ groups is presented in Figure 6C. The upregulation of the inflammatory response (19.7%)- and extracellular matrix (ECM) (17.9%)-related genes was the highest in the CCl_4_/saline group. The upregulation of the inflammatory response and ECM genes decreased after NK cell treatment. In addition, inflammatory response (2.5%)- and ECM (1.4%)-related genes were downregulated in the CCl_4_ + NK/CCl_4_ group. Since inflammatory response and ECM play an important role in cirrhosis, we examined these two categories of genes in the DEGs. In 1532 DEGs, 33 downregulated genes in the ECM (Table 2 and Table 3 left) and 41 downregulated genes in the inflammatory response (Table 2 and Table 3 right) were found.

### 3.6. qRT-PCR Verification of the Downregulated DEGs Related to ECM and Inflammatory Response

Expression patterns of functionally identified ECM and inflammatory response genes in downgraded DEGs were confirmed by qRT-PCR. The representative expression patterns are displayed in Figure 7A,B. QuantSeq 3′ mRNA sequencing expressed as a heatmap shows that the expression level of specific genes was significantly decreased in the CCl_4_ + NK group compared to the CCl_4_ group. Mainly, matrix metalloproteinases (MMPs) related to the degradation of the matrix and collagenases and TGF-β related to liver fibrosis formation were identified. Regulation-related (CCL19, S100a8, CCL1, etc.) and macrophage inflammatory response (CCL4, CD68, etc.) genes were identified among the inflammatory response genes. For verification of the identified genes, mRNA expression levels of 13 genes (MMP3, MMP11, MMP8, LAMB3, COL15a1, MMP12, ITGA6, COL8a2, TGFb3, MMP14, TGFb2, MMP9, LOXL1) related to the ECM and 14 genes (CSF1, CCL4, TNIP2, CCL2, FCGR1, CD68, CD14, SAA1, CCL19, ADAM8, ICAM1, ITGAM, S100a8, S100a9) related to the inflammatory response were analyzed using qRT-PCR (Figure 7C,D). The results of the qRT-PCR verification were consistent with QuantSeq 3′ mRNA sequencing. These results elucidated the therapeutic effect of NK cell therapy in the CCl_4_-induced liver cirrhosis mouse model at the genetic level.

Taking all the data together, our study on NK cell therapy in a CCl_4_-induced liver cirrhosis mouse model could be summarized as Figure 8.

## 4. Discussion

Previous studies have shown that NK cell functions are suppressed in progressive liver fibrosis [22,35]. Furthermore, NK cells were reported to have a protective effect against a CCl_4_-induced-liver cirrhosis mouse model [20,21,22]. We used the liver cirrhosis mouse model (Figure 2B) to evaluate the therapeutic potential of NK cell transplantation, which effectively reduced liver fibrotic lesions and cirrhosis progression (Figure 3B). We also examined whether NK cell transplantation could reduce inflammation, activated HSC number, and macrophage infiltration (Figure 4A–C). Finally, the localization of NK cells in the liver of the cirrhotic mice was visualized with BLI (Figure 5D,E).

Current therapeutics for liver cirrhosis can be summarized by several treatment options. One of the considerable therapeutic options is anti-hepatic fibrosis drugs [36]. In addition, gene therapy, including siRNA and miRNA and cell therapy, could be another therapeutic option for liver cirrhosis [36]. However, because of complicated mechanisms and differences in animal models and patients, drug therapy could lead to poor efficacy [36]. In addition, gene therapy has the strength of providing specificity, but it is essential to exclude systemic unwanted effects. The NK cell therapy is relatively free from these problems and has the advantage of being versatile for various other diseases if specific isolation/expansion is performed as we have proposed. For example, the accumulation of NK cells in the lung was elucidated in the CCl_4_-induced liver cirrhosis mouse model through the BLI images of Figure 5D,E, and it can also be applied to additional lung-related studies related to the CCl_4_-induced liver cirrhosis mouse model.

The therapeutic role of NK cells can only be examined by acquiring sufficient numbers of NK cells. The NK cell frequency in the spleen was reported to be only about 4% of total cells in 2–3-month-old C57BL/6J male mice [25]. Moreover, splenic NK cells were significantly reduced in aged mice (18–19 months old) compared to young mice (2–3 months old) [25]. Even with a nearly 100% acquisition of NK cells from the mouse spleen, multiple mice are needed to obtain about 1 × 10^7^ NK cells. Therefore, our NK cell isolation and expansion method, using fewer mice and allowing for the acquisition of a large number of NK cells for research and therapeutic applications, might be a better alternative. It was reported that there were differences in NK cell receptor expression and IFN-γ production in NK cells following IL-2 stimulation according to age and sex [25]. The aged male mice displayed an increased expression of NKG2D in the NK cells compared to young male mice, and young female mice produced significantly more IFN-γ than young male mice after 1000 U/mL IL-2 stimulation [25]. We speculated that the high expression of stress ligands by aged cells might enhance the expression of its corresponding NKG2D activation receptor in endogenous NK cells. In addition, since aged mice have low NK cell frequency, NK cell supplementation might be beneficial in the aged and diseased liver.

The NK cells that were isolated from the spleen and expanded were mostly CD3e-NKp46+ (>98%) (Figure 1B). Our method showed that just one week of incubation could expand NK cells without significant viability differences (Figure 1C). Moreover, the NK cells exhibited effector functions with a high expression of activation surface markers and a high level of IFN-γ secretion (Figure 1B,D). These results indicated that the NK cells were functional and had typical characteristics of NK cells.

Tracking the location of the immune cells at the target site can provide important information helpful in elucidating the role of immune cells. We employed a reporter gene imaging technique, BLI, to track NK cells in a cirrhosis mouse model [26]. The transgenic mouse expressing codon-optimized luciferase [28] as a source of NK cells was used to obtain more sensitive luminescence images. The visualization of the NK cells in the cirrhotic liver was important in our study, providing additional evidence for the role of NK cells in the cirrhotic liver. Relatively low signals were observed in the liver after 36 h of NK cell inoculation in the control mice, but high signals were observed in the CCl_4_-induced cirrhotic mice 36 h after NK cell inoculation (Figure 5D,E). Besides hepatic signals, high pulmonary signals were simultaneously visualized, requiring a further investigation of their relationship with the lungs (Figure 5E). Accumulation in other organs, such as the heart, kidney, and muscle tissues, was low in both groups (Figure 5D,E).

A recent study reported that CXCL12 paracrine signaling induced liver metastasis by silencing NK cells in activated HSCs in a liver metastasis model of subcutaneous breast cancer [37]. Thus, the interaction between NK cells and HSCs appears to be important in the cancer environment and in the liver cirrhosis model in this study. Therefore, the relationship between the reduction of cirrhosis and NK cells and the specific mechanisms of decreased activated HSC number and the blockade of macrophage infiltration requires further study and is necessary to elucidate the NK cell therapeutic mechanism involved in the liver cirrhosis.

We performed an RNA transcriptome analysis of the liver tissue from the normal, induced cirrhosis, and NK-cell-treated groups after the induction of cirrhosis. By examining the DEGs in each group, differences were found in the ECM and inflammatory response-related genes (Figure 7A,B), which were verified by RT-PCR (Figure 7C,D). These results confirmed that the NK cell-mediated treatment effect on cirrhosis was related to the ECM and inflammation (Figure 7A–D). The expression levels of the subfamily of MMPs, collagen, and TGF-β associated with the fibrotic process were significantly decreased after NK cell treatment (Figure 7A,C).

Hepatocyte death has been shown to initiate the liver fibrosis pathway [38]. Cellular components released from dead hepatocytes induced HSC activation, proliferation, transformation, and collagen production, leading to fibrosis, cirrhosis, and cancer [39]. HSC activation is known to be induced by platelet-derived growth factor, interleukin (IL)-1, tumor necrosis factor (TNF)-α, and others. It has been reported that TNF and IL-17 stimulated the differentiation of hematopoietic stem cells into myofibroblast-like cells and induced the expression of α-SMA and collagen synthesis through the arginase–proline pathway [40,41,42].

NK cells are activated by recognizing virus invasion and cancer cells, and they induce immune responses by producing large amounts of TNF-α and IFN-γ. CD4+ T helper 1 (T_h1_) cells reduce TGF-β-induced fibrosis-promoting activity through IFN-γ secretion and reduce collagen production by suppressing arginase function via the IFN-γ-induced JAK/STAT pathway, alleviating fibrosis [21,41,43,44]. It was reported that exosomes derived from NK cells inhibited HSC activation and CCl_4_-induced hepatic fibrosis [45]. In addition, a recent study reported that the metabotropic glutamate receptor 5 (mGluR5) of NK cells increased IFN-γ secretion through the MEK/ERK pathway, suggesting additional cytotoxicity to activated HSCs [46]. We found that the IFN-γ secretion increased 7.5 times after one week of NK cell expansion, compared to before the expansion (Figure 1D). Taken together, our data suggested that the secretion of IFN-γ by NK cells was involved in the alleviation of fibrotic liver pathology.

Furthermore, the mechanism by which NK cells directly kill HSCs might also help alleviate fibrosis. NK cells recognize their target cells through receptors, and stimulatory and inhibitory receptors determine NK cell activation status and their target cell killing effects. The mechanisms of the NK-cell-mediated killing of HSCs have been summarized in a review article [38]. The balance between stimulatory receptors, including NKG2D, NKp30, NKp36, NKp44, and NKp46, and inhibitory receptors, including killer cell immunoglobulin-like receptor (KIR) and Ly49, was reported to be involved in NK cell regulation [47].

In particular, NKG2D and NKp46 are involved in killing HSCs [22,48,49,50]. NKG2D ligands, including retinoic acid-induced early gene 1 (Rae1), MHC class I polypeptide-related sequence A (MICA), and UL16 binding protein (ULBP), are expressed on activated HSCs [22,48,49]. NK cells can kill activated HSCs by NKG2D- and tumor necrosis factor-related apoptosis-inducing ligand (TRAIL)-dependent mechanisms [49]. Besides TRAIL, other NK cell surface ligands, including the Fas ligand (FasL), can bind to receptors on HSCs and trigger HSC death through caspase activation and apoptosis [17]. NKp46 also plays a role in triggering lytic activity with its ligand NCR1 [50]. We obtained 61.55 ± 3.47% of NKp46+NKG2D+ NK cells using our expansion method (Figure 1B), indicating that the expansion method can retain the killing function of NK cells. In addition, NK cell-mediated HSC cytotoxicity could be induced by the secretion of granules containing perforin and granzyme A/B, which induce HSC apoptosis [18]. As shown in Figure 1B, we identified NKp46+ NK cells (>99%) and NKp46+NKG2D+ NK (61.55 ± 3.47%) cells in the NK cells. The relationship between cirrhosis and specific subsets of NK cells, such as NKG2D+ NK cells, needs to be further investigated by various methods, including single-cell genomics [51], and might increase the potential of specific cell-based therapeutic techniques [52].

QuantSeq 3′ mRNA sequencing revealed thirty-three downregulated genes associated with ECM after the NK cell treatment of CCl_4_-induced cirrhosis, including six MMP genes (MMP3, MMP8, MMP9, MMP11, MMP12, and MMP14) (Figure 7A,C). MMPs are a family of extracellular endopeptidases that are thought to be responsible for ECM degradation. For example, MMP2 (gelatinase A) is involved in HSC proliferation and the activation of TGF-β [53]. Additionally, MMP2, MMP3, and MMP9 are highly expressed during the acute phase of tissue damage [53], and the role of MMP12 in the treatment of mesenchymal stem cells for liver fibrosis has been reported [54]. Because each specific MMP can perform pro-fibrotic and anti-fibrotic dual functions, the relationship between specific MMPs and NK cell treatment needs to be further investigated.

In addition, we reported that lysyl oxidase-like-1 (LOXL1) was related to NK cell therapeutics in the CCl_4_-induced liver cirrhosis mouse model (Figure 7, Table 2). LOXL1 is one of the lysyl oxidase (LOX) family proteins. It was reported that LOXL1 was related to the progression of various tumors, such as glioma, gastric cancer, colorectal cancer, pancreatic ductal adenocarcinoma (PDAC) [55]. However, the association between LOXL1 and liver cancer or pathophysiology has not been well known; however, about a 30-fold increase in the mRNA expression level of LOXL1 has been reported in a CCl_4_-induced liver cirrhosis mouse model [56], but the detailed mechanism requires further investigation, and its relation with liver cancer is not well known. Considering the role of LOXL1 in cancer progression, it would be an interesting topic to investigate the role of LOXL1 in liver cancer and the possibility of NK cells’ therapeutic effects.

In summary, our results showed that NK cells could function similarly to the original NK cells and had protective effects in the CCl_4_-induced liver cirrhosis model. We provided evidence that NK cells could reduce cirrhosis progression by decreasing inflammation and reducing HSC and macrophage numbers in the cirrhotic liver. The therapeutic effects of NK cell transplantation in liver cirrhosis were related to ECM and inflammation. Our study suggests that NK cells can potentially be used for therapeutic applications.

## Figures and Tables

**Figure 1 biomedicines-11-01090-f001:**
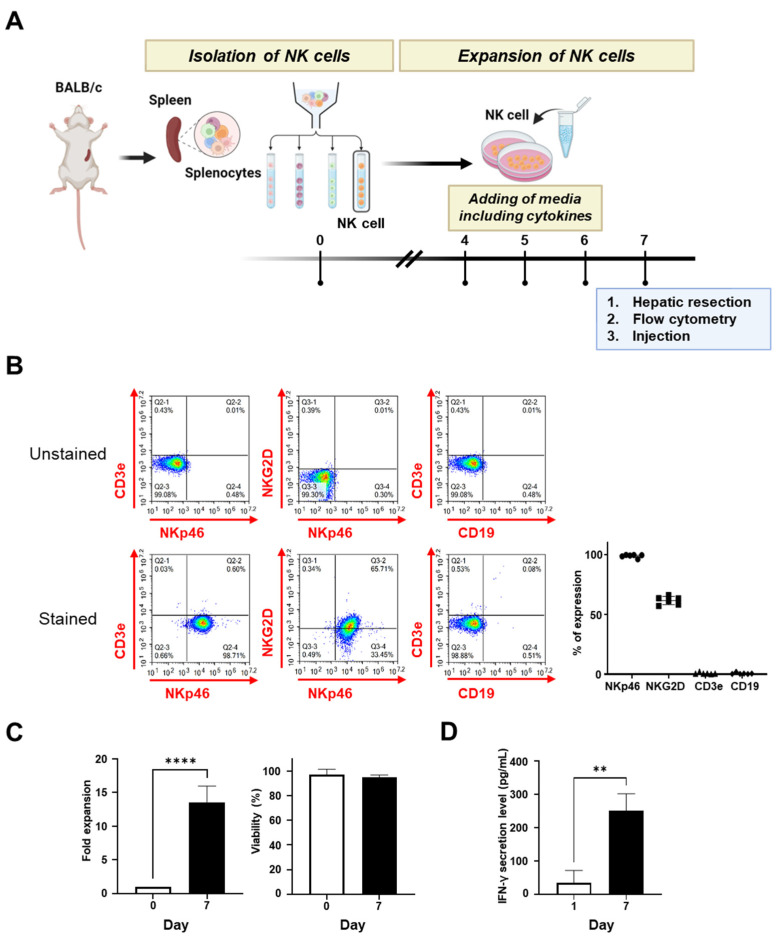
Evaluation of NK cell characterization after NK isolation and expansion. (**A**) Schematic time course of NK cell isolation and expansion methods. (**B**) Flow cytometric analysis of expanded population: NKp46+ NK cells (98.86 ± 1.39%) and NKp46+NKG2D+ NK cells (61.54 ± 3.47%). NKG2D, natural killer group 2, member D. (**C**) Comparison of the expanded population before (day 0) and after (day 7) isolation and expansion. (**D**) Comparison of IFN-γ secretion levels between day 1 (33.65 ± 38.12 pg/mL) and day 7 (251.86 ± 50.58 pg/mL). Data represent means ± SD. ** *p*-values < 0.005 and **** *p*-values < 0.0001. Figure was created with Biorender.com.

**Figure 2 biomedicines-11-01090-f002:**
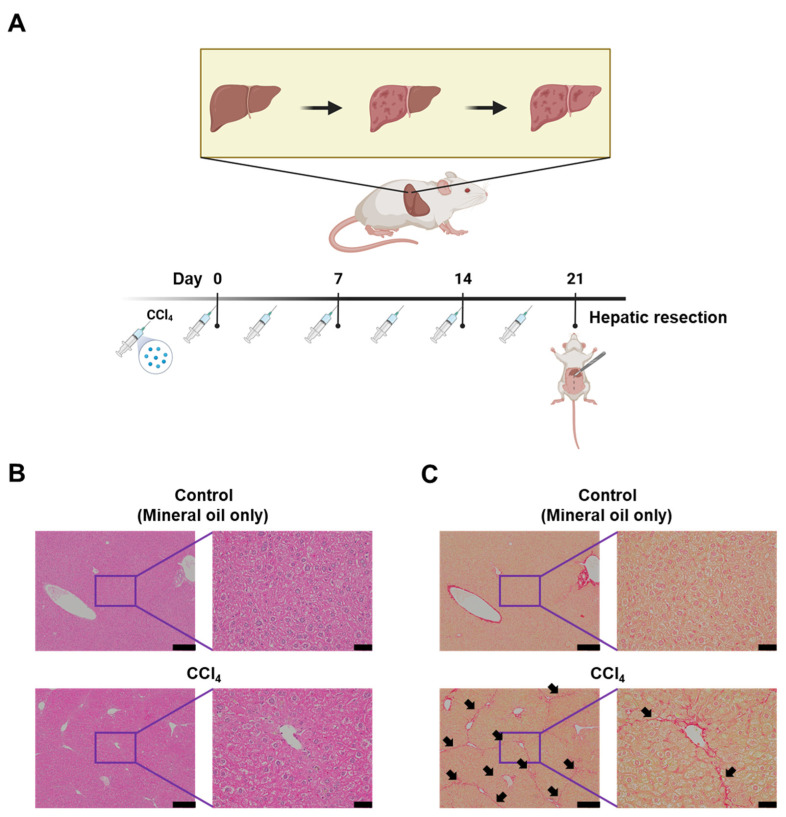
Evaluation of the CCl_4_-induced liver cirrhosis mouse model. (**A**) Schematic time course of the CCl_4_-induced liver cirrhosis mouse model. (**B**) Representative images of H&E staining in the control (mineral oil only) and CCl_4_-induced liver cirrhosis mouse model. (**C**) Representative images of Sirius red staining in the control (mineral oil only) and CCl_4_-induced liver cirrhosis mouse model. The image on the (**right**) was obtained from the (**left**) purple square area. Black arrow, pathological positive staining of fibrosis. Scale bar, (**left**), 200 μm, (**right**), 40 μm. Figure was created with Biorender.com.

**Figure 3 biomedicines-11-01090-f003:**
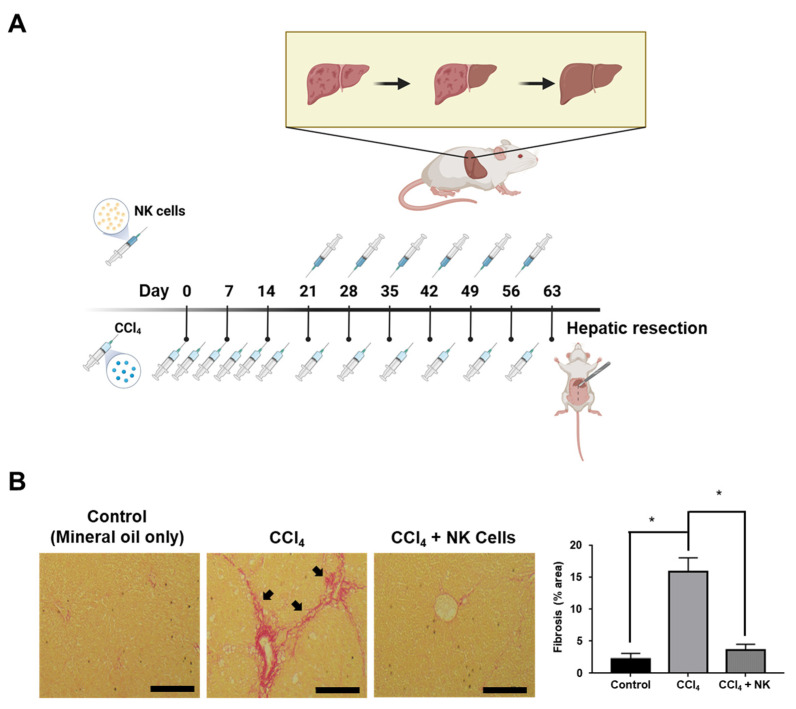
Decrease in liver cirrhosis progression after NK cell transplantation in the CCl_4_-induced liver cirrhosis mouse model. (**A**) Schematic time course of CCl_4_ treatment and injection of NK cells. (**B**) Representative images of Sirius red staining for histological visualization of collagen I and collagen III fiber level, compared across control (mineral oil only), CCl_4_-only treated, and CCl_4_ + NK cell-treated groups. Scale bar, 200 μm. Black arrow, pathological positive staining of fibrosis. (**Right**) graph demonstrates the quantitative analysis of the red fibrotic area. Data represent means ± SD. * *p*-values < 0.05. Figure was created with Biorender.com.

**Figure 4 biomedicines-11-01090-f004:**
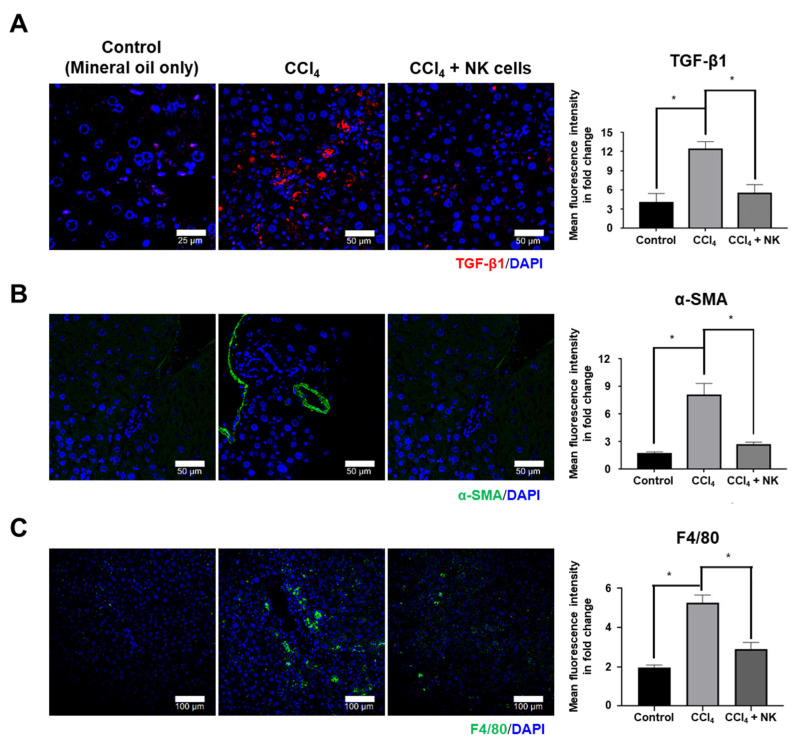
Decrease in fibrosis after NK cell transplantation in the CCl_4_-induced liver cirrhosis mouse model. (**A**) Representative TGF-β1 (pro-fibrogenic and inflammatory marker) immunofluorescence images and quantitative analysis. (**B**) Representative α-SMA (hepatic stellate activation marker) immunofluorescence images and quantitative analysis. (**C**) Representative F4/80 (macrophage surface marker) immunofluorescence images and quantitative analysis. Data represent means ± SD. * *p*-values < 0.05.

**Figure 5 biomedicines-11-01090-f005:**
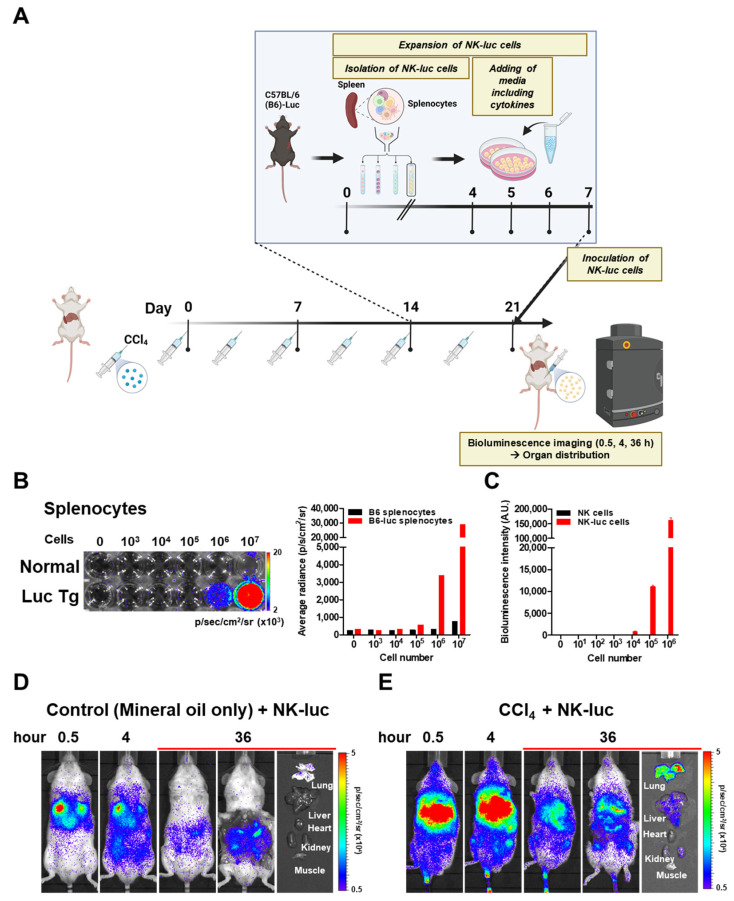
In vivo bioluminescence tracking of NK cells in the CCl_4_-induced liver cirrhosis mouse model. (**A**) Schematic time course of CCl_4_ treatment and BLI. (**B**,**C**) In vitro bioluminescence assay of splenocytes (**B**) and NK and NK-luc cells (**C**) from codon-optimized luciferase transgenic mice. Data represent means ± SD. (**D**,**E**) Representative BLI in control (mineral oil only) + NK-luc cell (**D**) and the CCl_4_ + NK-luc cell (**E**) treated mice (Imaging time: 0.5, 4, 36 h after NK-luc inoculation). After imaging at 36 h, mice were sacrificed, and organs were harvested (Control + NK-luc group n = 2, CCl_4_ + NK-luc group n = 3). Figure was created with Biorender.com.

**Figure 6 biomedicines-11-01090-f006:**
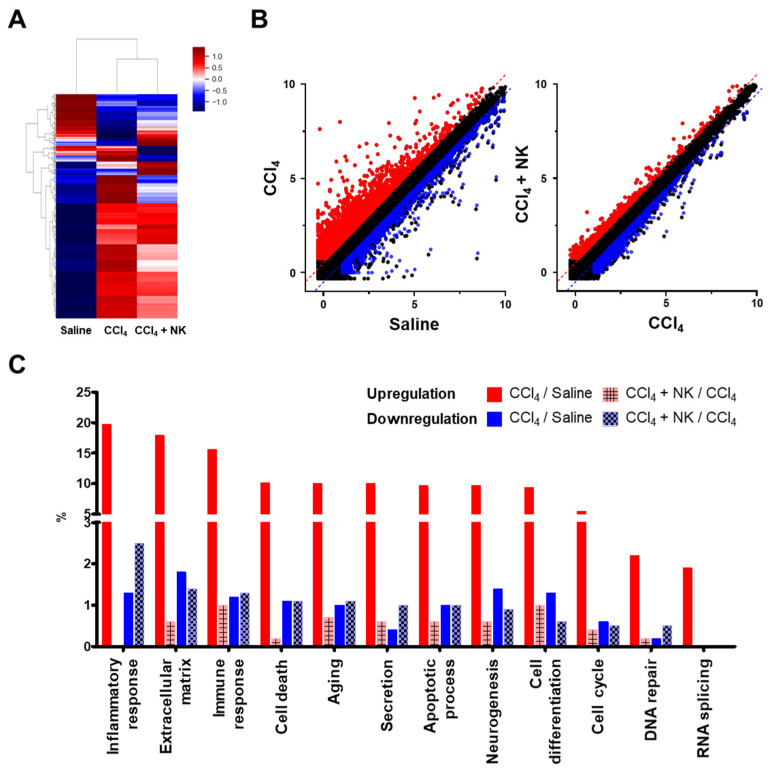
QuantSeq analysis of treatment-dependent DEGs. (**A**) Cluster heatmap of 1532 gene sets (fold-change = 1.5, *p*-value = 0.050, normalized data (log2 = 1)). The relative gene expression level was indicated by red for high expression and blue for low expression. (**B**) Scatter plot of DEGs. (**C**) Bar graph of gene ontology (GO). The bar indicates the GO terms related to gene functions and the percentage of differential expression in upregulated or downregulated genes in each category. Red and blue indicate >1.5-fold increased and decreased expression, respectively.

**Figure 7 biomedicines-11-01090-f007:**
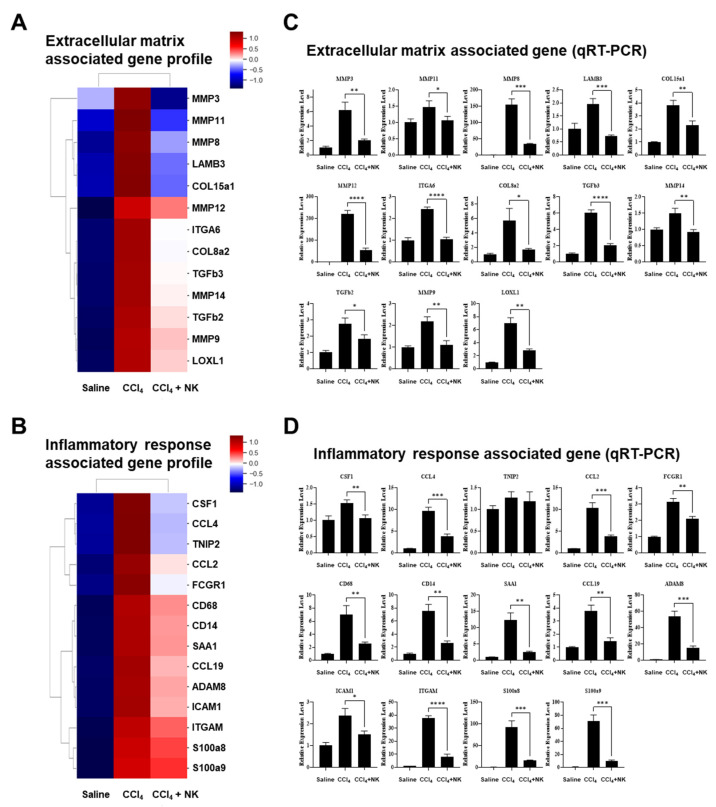
Expression levels of DEGs involved in extracellular matrix and inflammatory response. (**A**,**B**) Cluster heatmap of upregulated extracellular matrix-associated genes (n = 13) (**A**) and inflammatory response-associated genes (n = 14) (**B**). The relative gene expression level was indicated by red for high expression and blue for low expression. (**C**,**D**) Relative gene expression of extracellular matrix-associated genes (**C**) and inflammatory response-associated genes (**D**) by qRT-PCR analysis. Data are presented as mean ± SD from three independent experiments. * *p*-values < 0.05, ** *p*-values < 0.005, *** *p*-values < 0.001, and **** *p*-values < 0.0001.

**Figure 8 biomedicines-11-01090-f008:**
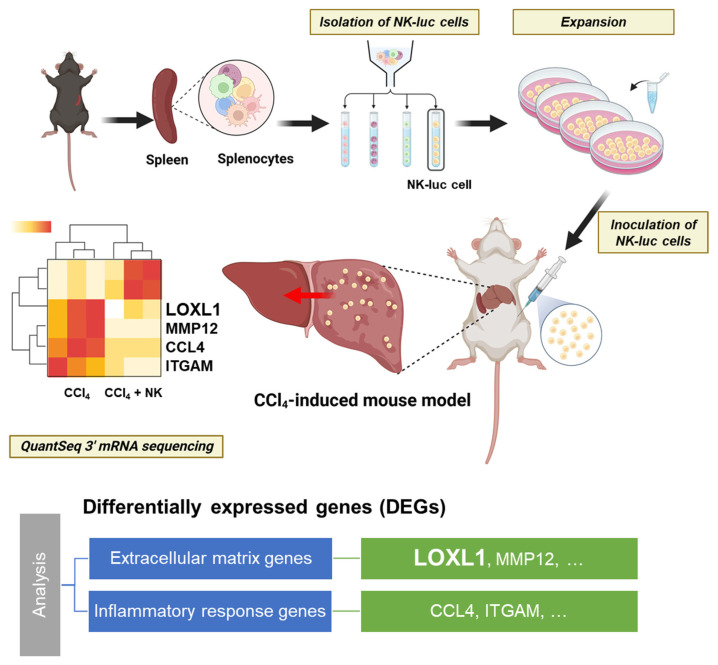
Summary of NK cell therapeutics in CCl_4_-induced liver cirrhosis mouse model in our study. Figure was created with Biorender.com.

**Table 1 biomedicines-11-01090-t001:** Target gene primer sequence.

Genes	Forward Sequence	Reverse Sequence	Pathways
mMMP3	CTCTGGAACCTGAGACATCACC	AGGAGTCCTGAGAGATTTGCGC	Extracellular matrix
mMMP8	GATGCTACTACCACACTCCGTG	TAAGCAGCCTGAAGACCGTTGG	
mMMP12	CACACTTCCCAGGAATCAAGCC	TTTGGTGACACGACGGAACAGG	
mMMP9	GCTGACTACGATAAGGACGGCA	TAGTGGTGCAGGCAGAGTAGGA	
mMMP14	GGATGGACACAGAGAACTTCGTG	CGAGAGGTAGTTCTGGGTTGAG	
mMMP11	GATTGATGCTGCCTTCCAGGATG	CAGCGGAAAGTATTGGCAGGCT	
mCOL15a1	ACACCCACAGTGACTCCCAAGA	TCCTCATTGCCCACGATGTCTC	
mCOL8a2	GAGTGTCCTCTGGCGGCGGA	AGTCCATTGGCAGCATCGGTAG	
mTGFb2	TTGTTGCCCTCCTACAGACTGG	GTAAAGAGGGCGAAGGCAGCAA	
mTGFb3	AAGCAGCGCTACATAGGTGGCA	GGCTGAAAGGTGTGACATGGAC	
mITGA6	CATCACGGCTTCTGTGGAGATC	CATTGTCGTCTCCACATCCTTCC	
mLAMB3	TGACCAGACCTATGGACACGTG	GTCACAGTGACCTCGTTGGCAT	
mLOXL1	CGACTATGACCTCCGAGTGCTA	GTAGTGGCTGAACTCGTCCATG	
mCCL4	ACCCTCCCACTTCCTGCTGTTT	CTGTCTGCCTCTTTTGGTCAGG	Inflammatory response
mS100a9	TGGTGGAAGCACAGTTGGCAAC	CAGCATCATACACTCCTCAAAGC	
mCD68	GGCGGTGGAATACAATGTGTCC	AGCAGGTCAAGGTGAACAGCTG	
mCSF1	GCCTCCTGTTCTACAAGTGGAAG	ACTGGCAGTTCCACCTGTCTGT	
mFCGR1	ACCTGAGTCACAGCGGCATCTA	TGACACGGATGCTCTCAGCACT	
mADAM8	TGCCAACGTGACACTGGAGAAC	GCAGACACCTTAGCCAGTCCAA	
mS100a8	CAAGGAAATCACCATGCCCTCTA	ACCATCGCAAGGAACTCCTCGA	
mSAA1	GGAGTCTGGGCTGCTGAGAAAA	TGTCTGTTGGCTTCCTGGTCAG	
mCCL2	GCTACAAGAGGATCACCAGCAG	GTCTGGACCCATTCCTTCTTGG	
mCD14	TTGAACCTCCGCAACGTGTCGT	CGCAGGAAAAGTTGAGCGAGTG	
mICAM1	AAACCAGACCCTGGAACTGCAC	GCCTGGCATTTCAGAGTCTGCT	
mTNIP2	AACCAGGAGCTGACAGCCATGA	CCAGCTCTTGAATCCTACTGTGC	
mITGAM	TACTTCGGGCAGTCTCTGAGTG	ATGGTTGCCTCCAGTCTCAGCA	
mCCL19	TCGTGAAAGCCTTCCGCTACCT	CAGTCTTCGGATGATGCGATCC	
mGAPDH	GTCTCCTCTGACTTCAACAGCG	ACCACCCTGTTGCTGTAGCCAA	Housekeeping

**Table 2 biomedicines-11-01090-t002:** Expression levels of DEGs involved in the extracellular matrix. The expression level of each gene is reported as the read count normalized to the log2 value. Red and blue background colors were used to visualize increased and decreased gene expressions, respectively, with the intensity of color representing the magnitude of change for fold-change values.

Gene Symbol	Fold Change		Normalized Data (log2)		
	CCl_4_/Saline	CCl_4_ + NK/CCl_4_	Saline	CCl_4_	CCl_4_ + NK
SERPINE1	12.519	0.139	1.378	5.024	2.178
THBS4	4.682	0.370	0.000	2.227	0.792
THBS1	3.378	0.373	1.079	2.835	1.413
CRISPLD2	5.071	0.459	1.299	3.641	2.516
MMP3	1.603	0.466	0.762	1.443	0.340
MMP12	15.310	0.494	0.638	4.575	3.557
ELAN	2.534	0.500	0.000	1.341	0.340
LTBP2	8.797	0.505	0.188	3.325	2.338
MMP8	2.836	0.505	0.430	1.935	0.949
BMPER	3.474	0.512	0.430	2.227	1.261
LOXL1	5.101	0.516	1.339	3.690	2.734
PLXDC2	5.697	0.516	0.875	3.385	2.430
COL15a1	2.356	0.524	2.501	3.737	2.804
OLFML2B	1.865	0.528	0.638	1.537	0.615
ANGPTL7	1.977	0.556	1.126	2.109	1.261
ADAMTS1	3.073	0.559	2.603	4.223	3.384
TGFb2	3.607	0.573	0.503	2.354	1.550
LRRC24	2.390	0.58	1.079	2.336	1.550
MGP	6.283	0.582	2.620	5.271	4.491
ITGA6	2.989	0.582	1.126	2.705	1.923
MFAP5	1.695	0.585	0.354	1.115	0.34
LABM3	2.051	0.588	1.031	2.067	1.300
VASN	1.718	0.59	3.625	4.405	3.644
TGFb3	2.952	0.596	1.031	2.593	1.845
MMP9	3.679	0.604	0.188	2.067	1.339
FMOD	3.765	0.611	0.820	2.732	2.021
WNT4	3.284	0.625	0.638	2.354	1.676
MMP14	2.711	0.627	5.089	6.527	5.854
MMP11	1.719	0.628	0.981	1.763	1.091
VWA1	1.663	0.646	0.572	1.306	0.677
IL16	2.404	0.647	1.453	2.719	2.090
LGALS3	23.571	0.651	1.525	6.084	5.465
COL8a2	2.336	0.656	0.430	1.654	1.045

**Table 3 biomedicines-11-01090-t003:** Expression levels of DEGs involved in the inflammatory response. The expression level of each gene is reported as the read count normalized to the log2 value. Red and blue background colors were used to visualize increased and decreased gene expressions, respec-tively, with the intensity of color representing the magnitude of change for fold-change values.

Gene Symbol	Fold Change		Normalized Data (log2)		
	CCl_4_/Saline	CCl_4_ + NK/CCl_4_	Saline	CCl_4_	CCl_4_ + NK
SAA2	18.764	0.318	3.786	8.016	6.363
NUPR1	5.942	0.344	0.188	2.759	1.220
CXCL2	6.373	0.361	0.820	3.492	2.021
THBS1	3.378	0.373	1.079	2.835	1.413
CCL2	5.937	0.427	0.701	3.271	2.045
SAA1	8.838	0.464	6.174	9.318	8.211
FPR2	6.102	0.474	1.453	4.063	2.985
CAMP	6.952	0.488	0.000	2.797	1.763
ADCYL	1.831	0.488	0.503	1.376	0.340
HPS1	1.708	0.490	3.191	3.964	2.935
CD14	7.534	0.491	1.525	4.438	3.412
ELANE	2.534	0.500	0.000	1.341	0.340
S100a8	33.124	0.507	0.981	6.031	5.050
GGT5	1.631	0.540	1.031	1.736	0.846
RELB	6.453	0.541	0.762	3.452	2.566
CXCL5	3.516	0.549	0.000	1.814	0.949
CCL21A	1.967	0.550	1.809	2.785	1.923
CSF1	2.565	0.559	2.351	3.710	2.871
CD68	5.757	0.560	2.446	4.972	4.134
PPARG	1.694	0.563	3.180	3.941	3.112
AIM2	2.621	0.566	0.981	2.371	1.550
CCRL2	4.28	0.566	0.273	2.371	1.550
HCK	6.707	0.573	0.762	3.507	2.705
PLA2G7	5.618	0.575	2.409	4.899	4.101
ADAM8	4.346	0.577	0.572	2.692	1.898
TICAM2	1.898	0.583	0.701	1.626	0.846
FCGR1	2.675	0.584	1.378	2.797	2.021
S100a9	28.555	0.592	1.453	6.289	5.533
CCL4	2.227	0.598	0.000	1.155	0.414
BCL6	2.036	0.599	5.327	6.353	5.613
SIGIRR	2.070	0.602	2.953	4.003	3.270
HDAC9	1.926	0.616	0.430	1.376	0.677
HYAL3	1.516	0.620	0.430	1.031	0.340
SMPDL3B	3.045	0.621	1.126	2.732	2.045
CCL19	3.173	0.628	0.097	1.763	1.091
MYO9A	6.399	0.635	0.638	3.316	2.660
ORM3	5.913	0.650	2.311	4.875	4.255
CX3CL1	2.860	0.657	0.572	2.088	1.483
ITGAM	5.006	0.661	0.572	2.896	2.300
TNIP2	1.921	0.663	2.185	3.126	2.533
ICAM1	2.885	0.664	2.518	4.047	3.456

## Data Availability

All raw data related to the results of this study can be obtained from the corresponding authors upon reasonable request.

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
