# Peer review of "Activated Natural Killer Cell Inoculation Alleviates Fibrotic Liver Pathology in a Carbon Tetrachloride-Induced Liver Cirrhosis Mouse Model"

_biomedicines, 2023, doi:10.3390/biomedicines11041090_

Round 1

Reviewer 1 Report

This article showed that activated NK cell inoculation alleviated fibrotic liver pathology in a CCl4-induced Liver Cirrhosis Mouse Model. There are some concerns in this manuscript that should  be corrected as follows:

1.    Title: It is preferable not to include abbreviations in the title; i.e. NK, CCL4.

2.    Abstract: The sentences "We examined the therapeutic effects of expanded NK cells in the carbon tetrachloride (CCl4)-induced liver cirrhosis mouse model " and "We investigated the therapeutic effect of activated NK  cells in the CCl4-induced liver cirrhosis mouse model " can be combined into one sentence.

3.    A conclusive statement should be added to the abstract.

4.    The novel points in this study should be clarified because there are previous studies that discussed a similar topic.

5.    The strain and the number of mice used in the present study should be mentioned in the beginning of "Materials & Methods" section.

6.    More details about the housing conditions of the animals used in this study should be added.

7.    How did the authors know that the animals were acclimatized?

8.     Page 3: A reference for the method of "Isolation and expansion of NK cells" should be added.

9.    The catalog numbers of the used kits should be added "e.g. Interferon-γ (IFN-γ) ELISA  kits".

10. A reference for the method of induction of liver cirrhosis by CCL4 should be added.

11. Page 5: More details about the methods of statistical analysis should be added.

12. Figure 2 and 3: Arrows that indicate the positive pathological findings should be added.

13. Page 9 Figure 4: The type of TGF-beta should be determined, whether it is type 1 or 2.

14. A collective diagram that summarizes the main findings of the present study should be added.

15. The manuscript should be checked regarding the grammatical and typing errors.

Author Response

This article showed that activated NK cell inoculation alleviated fibrotic liver pathology in a CCl4-induced Liver Cirrhosis Mouse Model. There are some concerns in this manuscript that should  be corrected as follows:

  1. Title: It is preferable not to include abbreviations in the title; i.e. NK, CCL4.

Response to the reviewer #1 : The abbreviation included in the title has been changed to full form (Line 2-4). We changed “NK cell” to “natural killer cell”, and “CCl4” to “carbon tetrachloride”. We also changed the tittle from “Activated NK cell Inoculation Alleviates Fibrotic Liver Pathology in a CCl4-induced Liver Cirrhosis Mouse Model” to “Activated Natural Killer Cell Inoculation Alleviates Fibrotic Liver Pathology in a Carbon Tetrachloride-induced Liver Cirrhosis Mouse Model”

  1. Abstract: The sentences "We examined the therapeutic effects of expanded NK cells in the carbon tetrachloride (CCl4)-induced liver cirrhosis mouse model " and "We investigated the therapeutic effect of activated NK cells in the CCl4-induced liver cirrhosis mouse model " can be combined into one sentence.

Response to reviewer #1: As you mentioned, two sentences were combined into one sentence and included in the manuscript (Line 19-20). Repetitive sentences have been removed and rewrote the abstract.

In this study, we examined the therapeutic effects of NK cells in the carbon tetrachloride (CCl4)-induced liver cirrhosis mouse model.”

  1. A conclusive statement should be added to the abstract.

Response to the reviewer #1 : Conclusive statement was added to the abstract (Line 32-35).

“Taken together, our research demonstrated that NK cells could have therapeutic effects in a CCl4-induced liver cirrhosis mouse model. In particular, it was elucidated that extracellular matrix genes and inflammatory response genes which were mainly affected after NK cell treatment, could be potential targets.”

  1. The novel points in this study should be clarified because there are previous studies that discussed a similar topic.

Response to the reviewer #1 : As you mentioned, there are many reports about the functional role of NK cell in the environment of fibrotic liver tissue, we revealed systematic targeting of NK cells in the cirrhotic liver in vivo and also analyzed changes of gene expression by RNA seq. Besides cirrhotic liver targeting, we observed the localization of NK cells in the lung (Figure 5D, and E). The RNA seq (Figure 6, 7) provided clues to reveal the mode of action for therapeutic effect of NK cells on the cirrhotic liver. Especially, we reported lysyl oxidase-like‑1 (LOXL1) was involved in the therapeutic effect of NK cells (Figure 7, Table 2). LOXL1 is one of the lysyl oxidase (LOX) family proteins and related to various tumor progression, such as glioma, gastric cancer, colorectal cancer, pancreatic ductal adenocarcinoma (PDAC) (Lin et al., 2021). However, the association between LOXL1 and liver cancer or pathophysiology has not been well known. The above description was included in our manuscript (Line 552-562).

In addition, we reported lysyl oxidase-like‑1 (LOXL1) was related to NK cell therapeutics in CCl4-induced liver cirrhosis mouse model (Figure 7, Table 2). LOXL1 is one of the lysyl oxidase (LOX) family proteins. It was reported that LOXL1 was related to various tumor progression, such as glioma, gastric cancer, colorectal cancer, pancreatic ductal adenocarcinoma (PDAC) [55]. However, the association between LOXL1 and liver cancer or pathophysiology has not been well known. Although, about 30-fold increase in the mRNA expression level of LOXL1 has been reported in a CCl4-induced liver cirrhosis mouse model [56], but the detailed mechanism requires further investigation, and the relation with liver cancer is not well known. Considering the role of LOXL1 in cancer progression, it would be an interesting topic to investigate the role of LOXL1 in liver cancer and the possibility of its NK cell therapeutic effects.”

  1. The strain and the number of mice used in the present study should be mentioned in the beginning of "Materials & Methods" section.

Response to the reviewer #1: We added the mouse strain and mouse number for each experiment (Line 116, Line 149, Line 186).

Line 116. Mice -> Seven-week-old male BALB/c mice (n=10)

Line 149. "Seven-week-old male BALB/c mice were administered intraperitoneally with mineral oil (control, n=4) or CCl4 (1.5 ml/kg, diluted 1:3 in mineral oil, n=4) twice a week for 3 weeks and once a week from 4th week (total 8 weeks) [32]."

Line 186. “For NK cell tracking experiments, five-month-old male C57BL/6 albino mice were administered mineral oil (control, n=2) or CCl4 (1.5 ml/kg, diluted in mineral oil, n=3) twice a week for 3 weeks.”

  1. More details about the housing conditions of the animals used in this study should be added.

Response to the review #1: The mouse housing condition was described in more detail than before (added humidity conditions and food types) (Line 111-112). “Mice were housed at room temperature with 40 to 60% humidity and fed standard chow diet and water ad libitum in 12-hour light/dark cycle.

  1. How did the authors know that the animals were acclimatized?

Response to the review #1: Referring to the previous report (Obernier and Baldwin, 2006), the mice used in the experiment went through an acclimatization period of at least one week. The experiments were performed after examining the sign of distress (e.g. unusual behavior) and the weight of the mice before the experiments. In addition, the above description was included in our manuscript (Line 112-113). “Mice were tested for experiments after at least one week of acclimatization.”

  1. Page 3: A reference for the method of "Isolation and expansion of NK cells" should be added.

Response to the review #1: We added a reference [31] in our manuscript (Line 120).

“NK cells were isolated from splenocytes by negative selection using an NK cell isolation kit (110-115-818, Miltenyi Biotec, Bergisch Gladbach, North Phine-Westphalia, Germany) according to the manufacturer’s instructions [31].”

  1. The catalog numbers of the used kits should be added "e.g. Interferon-γ (IFN-γ) ELISA kits".

Response to the review #1: We added the catalog number of the used kits in our manuscripts (Line 118, Line 138, Line 218).

Line 118. NK cells were isolated from splenocytes by negative selection using an NK cell isolation kit (110-115-818, Miltenyi Biotec, Bergisch Gladbach, North Phine-Westphalia, Germany)

Line 138. IFN-γ cytokine was measured by the enzyme-linked immunosorbent assay (ELISA) (SIF50C, R&D systems)

Line 218. First-strand cDNA was synthesized from 1 μg of total RNA using the TOPscript cDNA synthesis kit (EZ005S, Enzynomics, Daejeon, Korea)

  1. A reference for the method of induction of liver cirrhosis by CCl4 should be added.

Response to the review #1: We added the reference [32] in our manuscript (Line 150). “Seven-week-old male BALB/c mice were administered intraperitoneally with mineral oil (control, n=4) or CCl4 (1.5 ml/kg, diluted 1:3 in mineral oil, n=4) twice a week for 3 weeks and once a week from 4th week (total 8 weeks) [32].”

  1. Page 5: More details about the methods of statistical analysis should be added.

Response to the review #1: The statistical analysis method was described in more detail and included in the manuscript (Line 231-235). “Experiments were conducted in triplicate. All data are expressed as the mean ± standard deviation (SD). Statistical significance was analyzed by t-test using GraphPad Prism software (GraphPad Software Inc., San Diego, CA, USA). All p-values reported are two-sided and significance was set at P < 0.05. * p-values <0.05, ** p-values <0.005, *** p-values <0.001, and **** p-values <0.0001.”

  1. Figure 2 and 3: Arrows that indicate the positive pathological findings should be added.

Response to the review #1: We added pathological positive staining of fibrosis in Figure 2C and 3B.

  1. Page 9 Figure 4: The type of TGF-beta should be determined, whether it is type 1 or 2.

Response to the review #1: TGF-β1 was clarified in our figure and related manuscript (Figure 4, Line 308, Line 314, Line 320).

Figure 4, Line 309, Line 312, Line 318. TGF-β -> TGF-β1

  1. A collective diagram that summarizes the main findings of the present study should be added.

Response to the review #1: Thank you for your advice. We added a collective diagram summarizing main findings of our studies in Fig. 8.

  1. The manuscript should be checked regarding the grammatical and typing errors.

Response to the review #1: Thank you for your comments. Grammatical and typing errors were reviewed throughout the manuscript.

Reviewer 2 Report

This article studied the therapeutic value of activated NK cells in the CCl4-induced liver cirrhosis mouse model. The results covered NK cell insolation, therapeutic effect evaluation, in vivo localization tracking, transcriptomic analysis, and some tissue morphological observations. Overall, this article shows an interesting investigation and can be recommended for publication after some revisions.

1) Abstract is not well organized and not brief, although it covers all the results involved. Please reduce the redundancy and follow the standard procedures: a) Background; b) questions or purpose; c) experiments; d) results; e) conclusions; f) significance.

2) In the introduction (line 76-86), the authors introduced some imaging techniques. I suggest that authors can talk about the unique advantages of BLI among those techniques and why this uniqueness is good for the current study. The properties like noninvasive visualization, high sensitivity, and selectivity are not unique to BLI, but also can be seen in other techniques.     

3) The results in Figure 1B have some problems because they lack controls for each of these flow cytometric graphs. Without control graphs (blank groups), percentages in quadrants of these graphs (that are drawn by authors) are not conclusive, because these percentages can be adjusted to any value we want.

4) Presentation of figures needs some adjustments. For example, all the microscopic images should have scale bars, which cannot be seen clearly in Figure 2B/C, Figure 3B, and Figure 4A/B/C. In Figure 4, the label of y-axis in the bar graph should be ‘fluorescence intensity in fold change’ instead of ‘mean value’. No scale bars in micrographs.

5) In the result section, please indicate the rationale for using a certain amount of NK cells to test therapeutic effects in your mouse model.    

5) In discussion, please discuss current therapeutics for liver cirrhosis and show the advantages of cell therapy used in the current study.

Author Response

Response to reviewers’ comments

Reviewer #2

Comments and Suggestions for Authors

This article studied the therapeutic value of activated NK cells in the CCl4-induced liver cirrhosis mouse model. The results covered NK cell insolation, therapeutic effect evaluation, in vivo localization tracking, transcriptomic analysis, and some tissue morphological observations. Overall, this article shows an interesting investigation and can be recommended for publication after some revisions.

1) Abstract is not well organized and not brief, although it covers all the results involved. Please reduce the redundancy and follow the standard procedures: a) Background; b) questions or purpose; c) experiments; d) results; e) conclusions; f) significance.

Response to the review #2: As you mentioned, we revised our abstract by merging redundant and repetitive parts, rearranged it according to the procedure, included conclusive statement with significance of our experiments (Line 19-20, 22-25, 27-28, 30, 32-35).

2) In the introduction (line 76-86), the authors introduced some imaging techniques. I suggest that authors can talk about the unique advantages of BLI among those techniques and why this uniqueness is good for the current study. The properties like noninvasive visualization, high sensitivity, and selectivity are not unique to BLI, but also can be seen in other techniques.

Response to the review #2: Thank you for your thoughtful comment. Besides BLI, fluorescence, PET, SPECT, and MRI each have their own strengths and weaknesses. Only when used appropriately, effective cell tracking images can be obtained. However, the BLI imaging method using syngenic transfer that we used does not require an excitation laser like fluorescence without ectopic expression of fluorescent proteins. Therefore, luminescence is not disturbed by autofluorescence and attenuation is much less than fluorescence imaging.  In addition, in the case of PET or SPECT, which are nuclear medicine methods, radioactive isotope labeling is required for cell tracking, and depending on the conditions to which cells are exposed during the labeling process, validation of changes in cell properties may be necessary, especially in the case of sensitive immune cells. In the case of MRI, since labeling of a contrast agent such as iron oxide is also required, it is also necessary to validate that there is no change in cell characteristics during the labeling process.

In addition, the above description was included in our manuscript (Line 89-93).

“In the case of PET or SPECT, radioactive isotope labeling is required for cell tracking, vali-dation of changes in cell properties may be necessary during labeling process [27]. In the case of MRI, since labeling of a contrast agent such as iron oxide is also required, it is necessary to validate that there is no change in cell characteristics during the labeling process [26, 27].”

3) The results in Figure 1B have some problems because they lack controls for each of these flow cytometric graphs. Without control graphs (blank groups), percentages in quadrants of these graphs (that are drawn by authors) are not conclusive, because these percentages can be adjusted to any value we want.

Response to the review #2: As you commented, relative comparison with control/blank group is essential in FACS analysis. Control data (unstained) were added to the Figure 1B.

4) Presentation of figures needs some adjustments. For example, all the microscopic images should have scale bars, which cannot be seen clearly in Figure 2B/C, Figure 3B, and Figure 4A/B/C. In Figure 4, the label of y-axis in the bar graph should be ‘fluorescence intensity in fold change’ instead of ‘mean value’. No scale bars in micrographs.

Response to the review #2: We added a scale bar to all microscopic images (Figure 2B/C, 3B, 4A/B/C). Also, in Figure 4, the y-axis label was changed to “fluorescence intensity in fold change”.

5) In the result section, please indicate the rationale for using a certain amount of NK cells to test therapeutic effects in your mouse model.

Response to the review #2: From the previous papers, it was elucidated that NK cell adoptive transfer therapy in response to liver fibrosis or hepatocelluar carcinoma a number of 5x105 cells (Choi et al., 2012, Nakano et al., 2018, Masahiro et al., 2006). Compared to single administration in previous experiments (Choi et al., 2012, Nakano et al., 2018, Masahiro et al., 2006), NK cell therapy could be performed by considering the strength of our experiment that NK cell expansion (>10 times) could be performed after isolation of NK cells without a significant difference in viability (Figure 1B, C). Therefore, there was no difficulty in increasing the number of cells (1x106 cells/injection) and the number of treatments (6 times), a dramatic reduction in the progression of liver fibrosis could be elucidated as shown in Figure 3B.

5) In discussion, please discuss current therapeutics for liver cirrhosis and show the advantages of cell therapy used in the current study.

Response to the review #2: Thanks for your advice. Current therapeutics for liver cirrhosis could be summarized several treatment options. One of the considerable therapeutic options is anti-hepatic fibrosis drugs for anti-inflammation/protection of hepatocytes, inhibition of hepatic stellate cells activation/proliferation, and inhibition of ECM production/promotion of ECM degradation are one of the therapeutic options (Tan et al., 2021). Gene therapy including siRNA and miRNA and cell therapy could be other therapeutic options for liver cirrhosis (Tan et al., 2021). However, because of complicated mechanisms and difference of animal models and patients, drug therapy could lead to poor efficacy (Tan et al., 2021). In addition, gene therapy has the strength of providing specificity, but it is essential to exclude systemic unwanted effects. The NK cell therapy is relatively free from these problems and has the advantage of being versatility to various other diseases if specific isolation/expansion is performed as we have proposed. For example, accumulation of NK cell in lung was elucidated in the CCl4-induced liver cirrhosis mouse model through the BLI images of Figure 5D and E, it can also be applied to additional lung related studies due to CCl4 treatment.

In addition, the above description was included in this version of manuscript (Line 436-447).

“Current therapeutics for liver cirrhosis could be summarized several treatment options. One of the considerable therapeutic options is anti-hepatic fibrosis drugs [36]. In addition, gene therapy including siRNA and miRNA and cell therapy could be other therapeutic options for liver cirrhosis [36]. However, because of complicated mechanisms and difference of animal models and patients, drug therapy could lead to poor efficacy [36]. In addition, gene therapy has the strength of providing specificity, but it is essential to exclude systemic unwanted effects. The NK cell therapy is relatively free from these prob-lems and has the advantage of being versatility to various other diseases if specific isola-tion/expansion is performed as we have proposed. For example, accumulation of NK cell in lung was elucidated in the CCl4-induced liver cirrhosis mouse model through the BLI images of Figure 5D and E, it can also be applied to additional lung related studies related to CCl4-induced liver cirrhosis mouse model.”
